# Adversarially-learned Inference via an Ensemble of Discrete Undirected Graphical Models

**Adarsh K. Jeewajee**
MIT CSAIL
jaks19@mit.edu

**Leslie P. Kaelbling**
MIT CSAIL
lpk@csail.mit.edu

## Abstract

Undirected graphical models are compact representations of joint probability distributions over random variables. To solve inference tasks of interest, graphical models of arbitrary topology can be trained using empirical risk minimization. However, to solve inference tasks that were not seen during training, these models (EGMs) often need to be re-trained. Instead, we propose an inference-agnostic adversarial training framework which produces an infinitely-large ensemble of graphical models (AGMs). The ensemble is optimized to generate data within the GAN framework, and inference is performed using a finite subset of these models. AGMs perform comparably with EGMs on inference tasks that the latter were specifically optimized for. Most importantly, AGMs show significantly better generalization to unseen inference tasks compared to EGMs, as well as deep neural architectures like GibbsNet and VAEAC which allow arbitrary conditioning. Finally, AGMs allow fast data sampling, competitive with Gibbs sampling from EGMs.

## 1 Introduction

Probabilistic graphical models [Koller and Friedman, 2009, Murphy, 2012] are compact representations of joint probability distributions. We focus on *discrete pairwise undirected* graphical models, which represent the independence structure between pairs of random variables. Algorithms such as belief propagation allow for inference on these graphical models, with arbitrary choices of observed, query and hidden variables. When the graph topology is loopy, or when the structure is mis-specified, inference through belief propagation is approximate [Murphy et al., 2013].

A purely generative way to train such a model is to maximize the likelihood of training data (ML), under the probability distribution induced by the model. However, evaluating the gradient of this objective involves computing marginal probability distributions over the random variables. As these marginals are approximate in loopy graphs, the applicability of likelihood-trained models to discriminative tasks is diminished [Kulesza and Pereira, 2008]. In these tasks, the model is called upon to answer queries expressed compactly as $(X_{\mathcal{E}} = x_{\mathcal{E}}, X_{\mathcal{Q}}, X_{\mathcal{H}})$, where from a data point $(x_1, \ldots, x_N)$ sampled from a certain data distribution $\mathbb{P}$, we observe the values of a subset $\mathcal{E}$ of the indices, and have to predict the values at indices in $\mathcal{Q}$ from a discrete set $\mathcal{X}$, with the possibility of some hidden variable indices $\mathcal{H}$ which have to be marginalized over:

$$\arg\max_{x \in \mathcal{X}} \mathbb{P}(X_i = x | X_{\mathcal{E}} = x_{\mathcal{E}}), \forall i \in \mathcal{Q}. \tag{1}$$

A distribution over queries of this form will be referred to as an inference task. If the distribution over queries that the model will be called upon to answer is known a priori, then the model's performance can be improved by shaping the query distribution used at parameter estimation time, accordingly.

In degenerate tasks, $\mathcal{E}$, $\mathcal{Q}$ and $\mathcal{H}$ are fixed across queries. When this is the case *and* $\mathcal{H}$ is empty, we could use a Bayesian feed-forward neural network [Husmeier and Taylor, 1999] to model the

distribution in (1) and train it by backpropagation. The *empirical risk minimization of graphical models* (EGM) framework of Stoyanov et al. [2011] and Domke [2013] generalizes this gradient-descent-based parameter estimation idea to graphical models. Their framework allows retaining any given graphical model structure, and back-propagating through a differentiable inference procedure to obtain model parameters that facilitate the query-evaluation problem. EGM allows solving the most general form of problems expressed as (1), where $\mathcal{E}$, $\mathcal{Q}$ and $\mathcal{H}$ are allowed to vary. Information about this query distribution is used at training time to *sample* choices of evidence, query and hidden variable *indices* ($\mathcal{E}, \mathcal{Q}, \mathcal{H}$), as well the *observed values* $x_{\mathcal{E}}$ across data points. The whole imperfect system is then trained end-to-end through gradient propagation [Domke, 2010]. This approach improves the inference accuracy on this specific query distribution, by orders of magnitude compared to the ML approach. One significant drawback of the EGM approach is that the training procedure is tailored to one specific inference task. To solve a different inference task, the model often has to be completely re-trained (as we see in section 4).

Instead, we would like to learn discrete undirected graphical models which generalize over different or multi-modal inference tasks. Our *adversarially trained graphical model* (AGM) strategy is built on the GAN framework [Goodfellow et al., 2014]. It allows us to formulate a learning objective for our graphical models, aimed purely at optimizing the generation of samples from the model. No information about inference tasks is baked into this learning approach. Our only assumption during training is that the training and testing *data points* come from the same underlying distribution. Although our undirected graphical models need to be paired to a neural learner for the adversarial training, they are eventually detached from the learner, with an ensemble of parameterizations. When using one of the parameterizations, our graphical model is indistinguishable from one that was trained using alternative methods. We propose a mechanism for performing inference with the whole ensemble, which provides the desired generalization properties across inference tasks, improving over EGM performance. Our learning approach is essentially generative, but the ensemble of models increases the expressive power of the final model, making up for approximations in inference and model mis-specification which affected the ML approach discussed above.

In the next sections, we discuss related work (2) and introduce our adversarial training framework (3) for undirected graphical models. Our first experiment (4.1), shows that although undirected graphical models with empirical risk minimization (EGMs) are trained specifically for certain inference tasks, our adversarially-trained graphical models (AGMs) can perform comparatively, despite having never seen those tasks prior to training. The second experiment (4.2) is our main experiment which showcases the generalization capabilities of AGMs across unseen inference tasks on images. We also compare AGMs against state-of-the-art neural models GibbsNet and VAEAC which, like AGMs, were designed for arbitrary conditioning. In the last experiment (4.3), we show that the combination of AGMs and their neural learner provide a viable alternative for sampling from joint probability distributions in one shot, compared to Gibbs samplers defined on EGMs.

## 2  Related work

Our work combines *discrete*, *undirected* graphical models with the *GAN* framework for training. The graphical model is applied in *data space*, with the *belief propagation* algorithm used for *inference*, over an *ensemble of parameterizations*.

Combining an ensemble of models has been explored in classification [Bahler and Navarro, 2000] and unsupervised learning [Baruque, 2010]. Combined models may each be optimized for a piece of the problem space [Jacobs et al., 1991] or may be competing on the same problem [Freund and Schapire, 1999]. Linear and log-linear ensemble combinations like ours have been analyzed by Fumera and Roli [2005] and the closest work which uses the ensemble approach, by Antonucci et al. [2013], combines Bayesian networks for classification. Furthermore, our ensemble is obtained through the idea of one module learning to output the parameters of another module, which has its roots in meta learning. In Bertinetto et al. [2016] and Munkhdalai and Yu [2017], one neural network learns to assign weights to another network with and without the help of memory, respectively.

Using GANs to generate data from *discrete* distributions is an active area of research, including work of Fathony and Goela [2018], Dong and Yang [2018], and Camino et al. [2018], with applications in health [Choi et al., 2017] and quantum physics [Situ et al., 2018]. Undirected graphical models have been embedded into neural pipelines before. For instance, Zheng et al. [2015] use them as

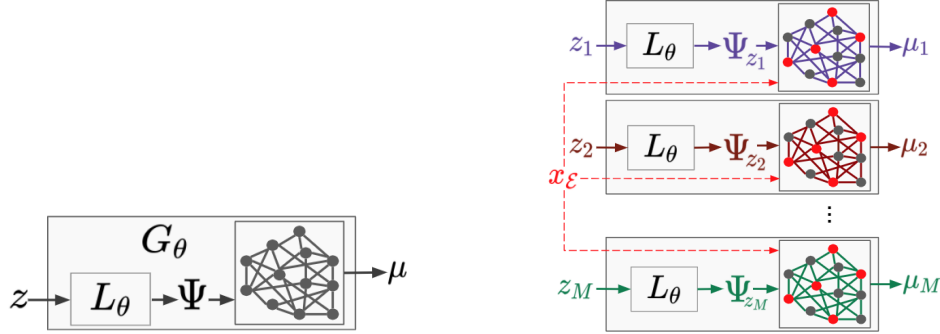

(a) **Training:** $L_\theta$ generates parameters $\Psi$ from $z \sim \mathcal{N}(0, I_m)$ for the graphical model. Belief propagation generates the vector $\mu$ from $\Psi$.

(b) **Testing/Inference:** Belief propagation on each of $M$ sampled models, given the *same* observations (red nodes), produces $M$ sets of conditional nodewise beliefs.

Figure 1: Our framework, during training (left) and during testing/inference (right).

RNNs, Ammar et al. [2014] and Johnson et al. [2016] use them in the neural autoencoder framework, Kuleshov and Ermon [2017] use them in neural variational inference pipelines, and Tompson et al. [2014] combine them with CNNs.

Other works use graph neural networks [Battaglia et al., 2018], but with some connection to classical undirected graphical models. For example, some works learn variants of, or improve on, message passing [Liu and Poulin, 2019, Satorras and Welling, 2020, Gilmer et al., 2017, Satorras et al., 2019]. Other works combine classical graphical models and graph neural networks with one another [Qu et al., 2019], while some use neural networks to replace classical graphical model inference entirely [Yoon et al., 2018, Zhang et al., 2019].

Some previous works combine graphical models with neural modules, but for *structured* inference. They assume a fixed set of input and output variables, solving problems such as image tagging [Chen et al., 2015, Tu and Gimpel, 2018] and classification [Tu and Gimpel, 2018]. These methods do *not* solve problems with arbitrary conditioning. On the other hand, *purely neural* models such as those developed in Ivanov et al. [2019], Douglas et al. [2017] and Belghazi et al. [2019] do tackle arbitrary conditioning problems formulated as (1). Being purely neural, the variational autoencoder (VAEAC) approach of Ivanov et al. [2019], however, requires masks defined over random variables during training to match conditioning patterns expected in test inference tasks, while we are *completely* agnostic to inference during training. We compare of our method against VAEAC in our experiments.

Among the work closest to ours, Fathony et al. [2018] learn *tractable* graphical models using *exact inference* through adversarial objectives. Chongxuan et al. [2018] and Karaletsos [2016] use graphical models in adversarial training pipelines, but to model posterior distributions. GANs have been used with graphs for high-dimensional representation learning [Wang et al., 2017], structure learning [Bojchevski et al., 2018] and classification [Zhong and Li, 2018]. Other relevant GAN works focus on inference in the data space without the undirected graphical structure. For example the conditional GAN [Mirza and Osindero, 2014] and its variants [Xu et al., 2019] allow inference, but conditioned on variables specified during training. [Donahue et al., 2016] and [Dumoulin et al., 2017] introduced the idea of learning the reverse mapping from data space back to latent space in GANs. GibbsNet [Lamb et al., 2017] is the closest model to us in terms of allowing arbitrary conditioning and being trained adversarially, although it is *not* a probabilistic model. The inference process of GibbsNet is iterative as it transitions from data space to latent space and back, stochastically several times, clamping observed values in the process. Our inference mechanism does not operate in a latent space, but is also iterative due to the belief propagation algorithm. Each model in our learned ensemble has significantly less parameters than GibbsNet, and we compare the performance of our method against GibbsNet in our experiments.

## 3   Method

**Preliminaries**    We aim to learn the parameters for pairwise discrete undirected graphical models, adversarially. These models are structured as graphs $G(V, E)$, with each node in their node set $V$

representing one variable in the joint probability distribution being modeled. The distribution is over variables $X_1^N := (X_1, \ldots, X_N)$. For simplicity, we assume that all random variables can take on values from the same discrete set $\mathcal{X}$.

A graphical model carries a parameter vector $\Psi$. On each edge $(i, j) \in E$, there is one scalar $\psi_{i,j}$ for every pair of values $(x_i, x_j)$ that the pair of connected random variables can admit. Therefore every edge carries $|\mathcal{X}|^2$ parameters, and in all, the graphical model $G(V, E)$ carries $k = |E||\mathcal{X}|^2$ total parameters, all contained in the vector $\Psi \in \mathbb{R}^k$.

Through its parameter set $\Psi$, the model summarizes the joint probability distribution over the random variables up to a normalization constant $\mathcal{Z}$ as:

$$q_{X_1^N}\left(x_1^N; \Psi\right) = \frac{1}{\mathcal{Z}} \prod_{(i,j) \in E} \psi_{i,j}\left(x_i, x_j\right). \tag{2}$$

Instead of incrementally updating *one* set of parameters $\Psi$ to train a graphical model $G(V, E)$, our method trains an uncountably infinite *ensemble* of graphical model parameters, adversarially. By learning an ensemble, we make up for the fact that our base graphical model structure may be mis-specified and may not be able to represent the true joint probability distribution over random variables of interest, and that inference on graphical models of arbitrary topologies is approximate. In our framework, our model admits a random vector $z \in \mathbb{R}^m$ sampled from a standard multivariate Gaussian distribution as well as a deterministic transformation $L_\theta$, from $z$ to a graphical model parameter vector $\Psi_z = L_\theta(z) \in \mathbb{R}^k$, where $\theta$ is to be trained. Under our framework, the overall effective joint distribution over random variables $X_1^N$ can be summarized as

$$p_{X_1^N}\left(x_1^N\right) \quad = \quad \int_{z \in \mathbb{R}^m} p_Z(z) \, p_{X_1^N|Z}\left(x_1^N|z\right) dz \quad = \quad \int_{z \in \mathbb{R}^m} p_Z(z) \, q_{X_1^N}\left(x_1^N; L_\theta(z)\right) dz \tag{3}$$

Through adversarial training, we will learn to map random vectors $z \in \mathbb{R}^m$ to data samples. The only learnable component of this mapping is the transformation of $z \in \mathbb{R}^m$ to $\Psi_z \in \mathbb{R}^k$ through $L_\theta$. Given $\Psi_z$, the joint distribution $q_{X_1^N}\left(x_1^N; \Psi_z\right)$ is given in (2) and since the goal of adversarial training is to produce high-quality samples which are indistinguishable from real data through the lens of some discriminator, the training process is essentially priming each $\Psi_z = L_\theta(z)$ to specialize on a niche region of the domain of the true data distribution. From the point of view of Jacobs et al. [1991], we will have learnt 'local experts' $\Psi_z$, each specializing to a subset of the training distribution. The entire co-domain of $L_\theta$ is our ensemble of graphical model parameterizations.

Finally, we will use an `inference` procedure throughout our exposition. Computing exact marginal probabilities using (2) is intractable. Hence, whenever we are given a graphical model structure, one parameter vector $\Psi$ and some observations $x_\mathcal{E}$, we carry out a fixed number $t$ of belief propagation iterations through the `inference`$(x_\mathcal{E}, \Psi, t)$ procedure, to obtain one *approximate* marginal probability distribution $\mu_i$, conditioned on $x_\mathcal{E}$, for every $i \in V$. Note that the distributions $\mu_i$ for $i \in \mathcal{E}$ are degenerate distributions with all probability mass on the observed value of random variable $X_i$. In our work, we will use this `inference` procedure with $\mathcal{E} = \emptyset$ and $\mathcal{E} \neq \emptyset$, during the learning and inference phases, respectively.

**Adversarial training**    Our adversarial training framework follows Goodfellow et al. [2014]. The discriminator $D_w$ is tasked with distinguishing between real and fake samples in data space. Our $(L_\theta, G(V, E))$ pair constitutes our generator $G_\theta$ as seen in figure 1a. Fake samples are produced by our generator $G_\theta$, which as is standard, maps a given vector $z$ sampled from a standard multivariate Gaussian distribution, to samples $\tilde{x}$.

One layer of abstraction deeper, the generative process $G_\theta$ is composed of $L_\theta$ taking in random vector $z \in \mathbb{R}^m$ as input, and outputting a vector $v \in \mathbb{R}^k$. The graphical model receives $v$, runs `inference`$(x_\mathcal{E} = \emptyset, \Psi = v, t = t')$, for a pre-determined $t'$, and outputs a set of marginal probability distributions $\mu_i$ for $i \in V$. Note that the set $\mathcal{E}$ of observed variables is empty, since our training procedure is inference-agnostic.

In summary, the graphical model extends the computational process which generated $v$ from $z$, with the deterministic recurrent process of belief propagation on its structure $E$. Note that a one-to-one correspondence between entries of $v$ and graphical model parameters $\psi_{i,j}(x_i, x_j)$ has to be pre-determined for $L_\theta$ and $G(V, E)$ to interface with one another.

Instead of categorical sampling from the beliefs $\mu_i$ to get a generated sample for the GAN training [Hjelm et al., 2017, Jang et al., 2017], we follow the WGAN-GP method of Gulrajani et al. [2017] for training our discrete GAN. In their formulation, the fake data point $\tilde{x}$ is a concatenation of all the marginal probability distributions $\mu_i$, in some specific order. This means that true samples from the training data set have to be processed into a concatenation of the $\mathcal{X}$-dimensional one-hot encodings of the values they propose for every node, to meet the input specifications of the discriminator.

We optimize the WGAN-GP objective (4) with the gradient $\nabla_{x'} \|D_w(x')\|_2$ penalized at points $x' = \epsilon x + (1 - \epsilon)\tilde{x}$ which lie on the line between real samples $x$ and fake samples $\tilde{x}$. This regularizer is a tractable 1-Lipschitz enforcer on the discriminator function, which stabilizes the WGAN-GP training procedure:

$$\min_w \max_\theta \mathop{\mathbb{E}}_{\tilde{x} \sim \mathbb{Q}} \left[ D_w(\tilde{x}) \right] - \mathop{\mathbb{E}}_{x \sim \mathbb{P}} \left[ D_w(x) \right] + \lambda \mathop{\mathbb{E}}_{x' \sim \mathbb{P}'} \left[ \left( \nabla_{x'} \|D_w(x')\|_2 - 1 \right)^2 \right]. \tag{4}$$

**Inference using the ensemble of graphical models**   Out of the various ways to coordinate responses from our ensemble of graphical model parameters (see section 2), we choose the log-linear pooling method of [Antonucci et al., 2013], for its simplicity. Given a query of the form $(X_\mathcal{E} = x_\mathcal{E}, X_\mathcal{Q}, X_\mathcal{H})$ as seen in (1), we call upon a finite subset of our infinite ensemble of graphical models. We randomly sample $M$ random vectors $z_1, \ldots, z_M$ from the standard multivariate Gaussian distribution and map them to a collection of $M$ parameter vectors $(\Psi_1 = L_\theta(z_1), \ldots, \Psi_M = L_\theta(z_M))$. $M$ sets of beliefs, for every node, are fetched through $M$ parallel calls to the `inference` procedure: `inference`$(x_\mathcal{E}, \Psi = L_\Theta(z_j), t = t')$ for $j = 1, \ldots, M$. The idea behind log-linear pooling is to aggregate the opinion of every model in this finite collection. Concretely, for every random variable $X_i$, its $M$ obtained marginal distributions $\mu_i(\cdot | x_\mathcal{E}; \Psi_j)$ for $j = 1, \ldots M$ are aggregated as we show in (5):

$$\hat{x}_i \quad = \quad \arg\max_{x \in \mathcal{X}} \prod_{j=1}^{M} \mu_i(x | x_\mathcal{E}; \Psi_j)^{\frac{1}{M}}. \tag{5}$$

Obtaining marginal probability distributions through an ensemble of graphical models is reminiscent of the tree-reweighted belief propagation algorithm of Wainwright et al. [2002], which produces these marginals using *one* set of parameters, and by re-formulating the belief propagation procedure to encompass the entire *polytope of spanning tree structures* associated with a set of random variables. In our work we learn a *variety of parameterizations*, over *one* fixed arbitrary graph topology, to make up for inaccuracies in that topology as well as the approximate nature of our inference procedure. Combining both ideas to learn an ensemble of parameterizations defined over a collection of structures is an interesting direction for future work.

## 4   Experiments

For inference tasks of the type formulated in (1), we need to define strategies for creating queries of the form: $(X_\mathcal{E} = x_\mathcal{E}, X_\mathcal{Q}, X_\mathcal{H})$. To construct any query, we start by sampling one data point from a distribution (data set) of interest. We then choose which variables to reveal as observations and we keep the original values of the rest of the variables (query variables) as targets. An inference task is created by applying one of the following (possibly stochastic) patterns to sampled data points:

(i) `fractional`$(f)$: A fraction $f$ of all variables is selected at random and turned into query variables, and the rest are revealed as evidence.

(ii) `corrupt`$(c)$: Every variable is independently switched, with probability $c$, to another value picked uniformly at random, from its discrete support. Then `fractional`$(0.5)$ is applied to to the data point to obtain the query as in (i).

(iii) `window`$(w)$: [Image only] The center square of width $w$ pixels is hidden and those pixels become query variables, while the pixels around the square are revealed as evidence.

(iv) `quadrant`$(q)$: [Image only] One of the four quadrants of pixels is hidden, and those pixels become query variables. The other three quadrants are revealed as evidence.

Some instantiations of these schemes, with specific parameters, on image data, are shown in figure 2. We note that the train and test query creation policies do not have to match. In fact, the strength of

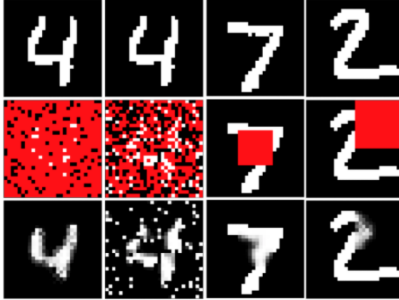

Figure 2: In columns 1 to 4, query-creation schemes are: `fractional(0.85)`, `corrupt(0.2)`, `window(10)` and `quadrant(1)`, respectively. Row 1: original data; row 2: data is converted to queries ($X_{\mathcal{E}} = x_{\mathcal{E}}, X_{\mathcal{Q}}, X_{\mathcal{H}}$) (non-red pixels are observations $x_{\mathcal{E}}$, red pixels are variables $X_{\mathcal{Q}}$ to be guessed); row 3: marginals produced by *one* AGM, where $\mathbb{P}(\text{pixel} = 1)$ is plotted. Note that the dots apparent in row 3 column 2 are due to unchanged corrupted pixels which we count as observations.

Table 1: Data set information and accuracies from experiments I and III. Every value is *averaged over 5 repeats*, with an error bar smaller than $\pm 0.5$.

| Name | Experiment I | | | Experiment III | | |
|---|---|---|---|---|---|---|
| | EGM | AGM (ours) | EGM-AGM | AGM sampler | Gibbs sampler (burn=0) | Gibbs sampler (burn=10) |
| **NLTCS** | 81.7 | 79.5 | +2.2 | 81.2 | 77.7 | 79.8 |
| **Jester** | 70.6 | 65.7 | +4.9 | 69.88 | 63.9 | 67.0 |
| **Netflix** | 66.0 | 63.9 | +2.1 | 64.7 | 61.4 | 61.8 |
| **Accidents** | 83.3 | 83.2 | +0.1 | 83.0 | 81.2 | 82.8 |
| **Mushrooms** | 88.0 | 87.3 | +0.7 | 86.8 | 85.3 | 86.1 |
| **Adult** | 92.1 | 92.1 | 0 | 92.0 | 90.6 | 91.9 |
| **Connect 4** | 88.8 | 88.6 | +0.2 | 88.4 | 85.9 | 88.4 |
| **Pumsb-star** | 86.7 | 82.2 | +4.5 | 84.1 | 77.7 | 81.8 |
| **20 NewsGroup** | 96.0 | 96.0 | 0 | 94.7 | 90.6 | 94.7 |
| **Voting** | 99.8 | 92.4 | +7.4 | 94.8 | 67.6 | 64.1 |
| **MNIST** | 93.8 | 94.5 | -0.7 | 93.0 | 91.0 | 92.8 |
| **Caltech-101** | 93.2 | 94.4 | -1.2 | 92.2 | 92.0 | 92.1 |
| **Stanford** | 91.5 | 91.6 | -0.1 | 69.0 | 68.4 | 68.4 |
| **SVHN** | 86.7 | 91.3 | -4.6 | 92.2 | 92.0 | 92.1 |

AGMs is their generalization capabilities to unseen inference tasks, and experiment II is designed to compare the performance of AGMs against other models when this mismatch occurs.

Concerning data sets, we use: **ten binary data sets** used in previous probabilistic modelling work (example Gens and Pedro [2013]) spanning 16 to 1359 variables, **two binarized image data sets** (28x28 pixels) which are `MNIST` [LeCun and Cortes, 2010] with digits 0 to 9, and `Caltech-101` Silhouettes [Li et al., 2004] with shape outlines from 101 different categories, **one discrete image data set** (30x40 pixels) made of downsampled segmentation masks from the `Stanford` background dataset, and **one RGB image data set** (32x32 pixels) which is the `SVHN` dataset with digits of house numbers (see appendix for how we encode continuous RGB values through node marginals).

## 4.1 Experiment I: Benchmarking

In this experiment, we compare the performance of AGMs against EGMs, on an inference task that the EGMs were specifically optimized for. While we expect EGMs to have slightly superior performance due to this unfair advantage, we observe that AGMs nevertheless show comparable performance.

We calibrate our models on each training data set separately, and test on 1000 unseen points. The inference task `fractional`(0.7) is used to test every model. EGMs train by minimizing the conditional log likelihood score under the inference task given by `fractional`(0.5). Accuracies are given in table 1 as the percentage of query variables correctly guessed, over all queries formed from the test set. We use identical randomized edge sets of size $5|\mathcal{V}|$ for non-image data, while a grid graph structure is used with image data (last four rows of table 1).

Results in table 1 show that AGMs trail EGMs on tasks with randomized graphs by a mean EGM-AGM difference of only 2.1, and AGMs surpass EGMs on all image data sets, with a mean EGM-AGM difference of -1.65, despite having never seen these inference tasks during training.

## 4.2 Experiment II: Generalization across inference tasks on images

Experiment I showed that an AGM can be used for inference on `fractional` tasks. In this experiment, we test if an AGM can generalize to other inference tasks, such as `corrupt`, `window` and `quadrant`, despite its inference-agnostic learning style. We also compare against other models which are built for arbitrary conditioning, similar to AGMs. On one hand, we would like to see if an EGM and a VAEAC (described in section 2), both of which assume specific inference tasks during training, generalize to query distributions that they were not exposed to. At the same time, we would like to see how a deep neural architecture like GibbsNet (described in section 2) compares to AGMs, as both of them are trained in an inference-agnostic and adversarial manner. Both VAEAC and GibbsNet follow the architectures given in their original papers.

In this experiment, every candidate model will be *separately* evaluated on `fractional`(0.5), `window`(7), `corrupt`(0.5) and `quadrant`(1) tasks. We train *one AGM* and *one GibbsNet architecture*, adversarially, by definition. For EGM and VAEAC, since they assume specific inference tasks during training, we will train:

- *one* EGM and *one* VAEAC by sampling queries successively from *all* inference tasks (MIX scheme)

- *multiple* EGMs and VAEACs, by sampling queries successively from all inference tasks, *except the specific task they are tested on* (MIX-1 scheme).

Tables 2, 3, 4 and 5 show the performances of these models, trained under different schemes (where applicable) and tested on tasks, spread horizontally. The most important metric is indicated in bold, and is the best result, per inference task (per column), obtained by comparing: EGM (MIX-1), VAEAC (MIX-1), GibbsNet and AGM. The MIX-1 scores, when compared to MIX scores, indicate how well EGMs and VAEACs perform on tasks that they have never seen during training. For both VAEACs and EGMs, their performances degrade drastically as soon as they face unseen tasks (seen by the difference between their MIX and MIX-1 rows). This shows that these models do not generalize well to unseen tasks. Out of all the models (EGM (MIX-1), VAEAC (MIX-1), GibbsNet and AGM) whose training procedures were agnostic to the evaluation task of each column in the result tables, AGMs perform the best. AGMs have the highest mean scores (calculated over all inference tasks), for all data sets, out of these models. If a practitioner is confident that their model will *never* have to face unseen inference tasks, then VAEACs may be the best choice for them, given that under the MIX scheme, these models obtain the highest results in most columns.

An interesting observation is that the `corrupt`(0.5) task causes the biggest drop in performances from MIX to MIX-1 schemes for EGMs and VAEACs, and proves to be hard for GibbsNet as well, as the latter learns a latent representation from data, and training data does not have corrupted pixels. AGMs are able to make up for the lack of exposure to corruption through its numerous parameterizations which are specialized to different parts of the data space, compared to the single parameterization of GibbsNet which is bound to be more general. Another interesting result emerges when EGMs are viewed in isolation, and when we compare results obtained on the `fractional`(0.5) task from experiment I to the results obtained on the same task, in experiment II, under the MIX setting. It is clear that preparing discrete undirected graphical models by training them on mixtures of tasks takes away from the specialization obtained when trained on *one* task specifically. Hence, for practitioners who wish to use these models, it is difficult to design a training curriculum that will ensure that the models perform consistently well across tasks. AGMs do not require any extra work

Table 2: Cross-task results on `MNIST` data. Per column, the bold value is best result between EGM (MIX-1), VAEAC (MIX-1), GibbsNet and AGM, and the shaded result is the best vertically.

| Model | Trained on | *Tested on* | | | | |
|---|---|---|---|---|---|---|
| | | f=0.5 | w=7 | c=0.5 | q=1 | mean |
| EGM | MIX | 93.6 ±0.2 | 66.3±0.2 | 82.6±0.2 | 86.9±0.3 | 82.4 |
| | MIX-1 | 87.4±0.1 | 64.1±0.3 | 68.2±0.1 | 84.2±0.1 | 76.0 |
| VAEAC | MIX | 94.2±0.4 | 72.4±0.4 | 79.8±0.4 | 87.9±0.3 | 83.6 |
| | MIX-1 | 85.5±0.4 | 61.2±0.5 | 65.1±0.4 | 81.3±0.1 | 73.3 |
| GibbsNet | - | 88.6±0.1 | 70.5±0.2 | 68.0±0.1 | 87.1±0.1 | 78.6 |
| AGM (Ours) | - | **94.5 ± 0.1** | **72.3 ± 0.1** | **79.2 ± 0.2** | **87.4 ± 0.1** | **83.4** |

Table 3: Cross-task results on `Caltech-101` data. Per column, the bold value is best result between EGM (MIX-1), VAEAC (MIX-1), GibbsNet and AGM, and the shaded result is the best vertically.

| Model | Trained on | *Tested on* | | | | |
|---|---|---|---|---|---|---|
| | | f=0.5 | w=7 | c=0.5 | q=1 | mean |
| EGM | MIX | 92.0 ±0.2 | 90.0±0.1 | 70.2±0.4 | 80.6±0.3 | 83.2 |
| | MIX-1 | 81.3±0.2 | 88.7±0.2 | 55.2±0.4 | 79.5±0.3 | 76.2 |
| VAEAC | MIX | 95.5±0.5 | 90.0±0.5 | 71.1±0.4 | 80.7±0.5 | 84.3 |
| | MIX-1 | 84.2±0.4 | 85.6±0.5 | 58.4±0.4 | 74.9±0.1 | 75.8 |
| GibbsNet | - | 89.4±0.1 | 90.0±0.3 | 57.0±0.4 | 78.2±0.1 | 78.4 |
| AGM (Ours) | - | **94.4 ± 0.3** | **94.2 ± 0.3** | **66.2 ± 0.5** | **80.1 ± 0.1** | **83.7** |

Table 4: Cross-task results on `Stanford` data. Per column, the bold value is best result between EGM (MIX-1), VAEAC (MIX-1), GibbsNet and AGM, and the shaded result is the best vertically.

| Model | Trained on | *Tested on* | | | | |
|---|---|---|---|---|---|---|
| | | f=0.5 | w=7 | c=0.5 | q=1 | mean |
| EGM | MIX | 89.2 ±0.5 | 68.9±0.4 | 82.1±0.5 | 83.1±0.5 | 80.8 |
| | MIX-1 | 84.1±0.2 | 57.8±0.3 | 71.8±0.2 | 78.3±0.2 | 73.0 |
| VAEAC | MIX | 89.7±0.3 | 76.3±0.3 | 82.0±0.3 | 83.9±0.5 | 83.0 |
| | MIX-1 | 82.1±0.3 | 70.6±0.5 | 60.4±0.4 | 72.1±0.2 | 71.3 |
| GibbsNet | - | 87.6±0.4 | **75.9 ± 0.5** | 61.2±0.4 | 78.4±0.4 | 75.8 |
| AGM (Ours) | - | **91.6 ± 0.1** | 75.4 ± 0.4 | **80.6 ± 0.5** | **82.3 ± 0.4** | **82.5** |

Table 5: Cross-task results on `SVHN` data. Per column, the bold value is best result between EGM (MIX-1), VAEAC (MIX-1), GibbsNet and AGM, and the shaded result is the best vertically.

| Model | Trained on | *Tested on* | | | | |
|---|---|---|---|---|---|---|
| | | f=0.5 | w=7 | c=0.5 | q=1 | mean |
| EGM | MIX | 91.4 ±0.5 | 81.9±0.4 | 74.1±0.5 | 91.6±0.5 | 84.8 |
| | MIX-1 | 85.9±0.5 | 75.3±0.3 | 60.7±0.4 | 84.2±0.4 | 76.5 |
| VAEAC | MIX | 91.8±0.4 | 80.0±0.4 | 73.1±0.5 | 92.5±0.5 | 84.4 |
| | MIX-1 | 88.7±0.5 | 64.4±0.5 | 70.6±0.5 | 87.0±0.5 | 77.7 |
| GibbsNet | - | 85.1±0.4 | 80.2±0.3 | 68.0±0.4 | 89.1±0.5 | 80.6 |
| AGM (Ours) | - | **91.3 ± 0.3** | **81.4 ± 0.3** | **77.8 ± 0.4** | **92.4 ± 0.5** | **85.7** |

from practitioners in terms of curriculum design, and AGM performances are close to performances of models which have already been exposed to evaluation tasks.

### 4.3 Experiment III: Sampling using AGMs instead of Gibbs sampling

Motivated by the crisp image samples generated from AGMs and smooth interpolations in latent space (see appendix), we decided to quantify and compare the quality of samples from AGMs, versus from Gibbs samplers defined on EGMs. If one wishes to use a graphical model principally for inference, they would have to make a choice between an EGM and an AGM. In this experiment, we show that AGMs provide added benefits, on top of generalizable inference. Namely, the learner-graphical model pair constitutes a sampler that produces high-quality samples in one shot (one pass from $z$ to $\Psi$ to $x$).

We would like some metric for measuring sample quality and we use the following, inspired by previous work on model distillation [Hinton et al., 2015]: given our two samplers, we will use data generated from them, and feed the data to newly-created models for training to solve an inference task, from scratch. The score attained by the new model will indicate the quality of the samples generated by the samplers.

Concretely, we train an AGM (A) and an EGM (B) on some training data set D from table 1. A is trained adversarially by definition, and B assumes the `fractional`$(0.5)$ inference task. We generate $1000$ samples from each model, and call these sampled data sets $S_1$ and $S_2$. If we now train a freshly-created EGM $E_1$ on $S_1$ and another one, $E_2$ on $S_2$, from scratch, then test them on the test data set corresponding to D, then which one out of of $E_1$ or $E_2$ has better performance, assuming everything else about them is identical? If $E_1$ performs better, then data from A was of better quality, else, Gibbs sampling on B produced better data. The inference task used to test $E_1$ or $E_2$ is `fractional`$(0.5)$.

For the Gibbs sampler defined on B, we try two scenarios: one where it uses no burn-in cycles to be similar to the one-shot sampling procedure of A, and one scenario where it has $10$ burn-in cycles. Interestingly, as seen in table 1, A is better than B regardless of the number of burn-in cycles, bar one exception, and performance when trained on data from A is not that far off the performance from real training data. For B, even 10 burnin steps are not enough for the Markov chain being sampled from to mix. The runtimes of both sampling procedures change linearly with the number of steps used (belief propagation steps with A and Gibbs sampling burnin with B), but since variables have to be sequentially sampled in Gibbs sampling, the process cannot be parallelized across nodes and edges of the graph, yielding poorer runtime, compared to belief propagation which is fully parallelizable and runs entirely through matrix operations on a GPU [Bixler and Huang, 2018] (see appendix for runtime analysis of our method).

In summary, sampling from an AGM is a viable tool and is an added benefit that comes with training AGMs. Our results also indicate that distilling the knowledge from our ensemble into single models may be a promising future line of work.

## 5 Conclusion

The common approach for training undirected graphical models when the test inference task is known a priori, is empirical risk minimization. In this work, we showed that models produced using this approach (EGMs) fail to generalize across tasks. We introduced an adversarial training framework for undirected graphical models, which instead of producing one model, learns an ensemble of parameterizations. The ensemble makes up for mis-specifications in the graphical model structure, and for the approximate nature of belief propagation on graphs of arbitrary topologies. As shown in our experiments, the inference-agnostic nature of our training method allows one AGM to generalize over an array of inference tasks, compared to an EGM which is task-specific. We also compared AGMs against deep neural architectures used for inference with arbitrary conditioning, such as GibbsNet and VAEACs, and AGMs were shown to perform better on unseen inference tasks. Finally, we showed that data can be sampled from AGMs in one shot, presenting an added benefit of using AGMs, and we illustrated that it is possible to distill the knowledge from our ensemble into single models.

## Broader Impact

Graphical models are interpretable models as they expose their independence structures. Such models provide transparency that would allow practitioners to understand biases that may have been imparted to the parameters. If one wishes to understand ways in which a graphical model is biased, one may condition on variables of interest and see exactly how the rest of the variables react.

The fact that our graphical model comes as an ensemble, makes it an editable model. For instance, if the model shows biases towards some characteristic of the data distribution being modeled, one can adapt our approach, and use a weighted recombination [Baruque, 2010] of the individual models in our ensemble, such that the models showing the undesirable bias are suppressed, or they can be completely zeroed out. This is not easy to do in black-box models such as neural networks.

As with most approaches, there are modes of the failure to our model, and the consequences would depend on the setting where the model is used, but the ability to perform arbitrary conditioning with our model can be seen as a safety feature. If our model is used in critical situations where certain combinations of variables have high importance, then our models can be queried with those combinations of variables and their response to such conditions can be well understood and improved, or made safer, if needed.

## Acknowledgments

We gratefully acknowledge support from NSF grant 1723381; from AFOSR grant FA9550-17-1-0165; from ONR grant N00014-18-1-2847; from the Honda Research Institute; from the MIT-IBM Watson AI Lab; and from SUTD Temasek Laboratories. Any opinions, findings, and conclusions or recommendations expressed in this material are those of the authors and do not necessarily reflect the views of our sponsors.

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
