[Reviews · NeurIPS 2020]

Review 1

Summary and Contributions: This work tackles the important topic of efficient generalization over inference tasks by presenting an inference agnostic training procedure for undirected graphical models. The authors propose AGM to adversarially train undirected graphical model by representing potentials as the output of a generator. Inference is done by sampling different set of potentials, carrying belief propagation and aggregating the obtained marginal distributions.

Strengths: The paper of this topic is important. Efficient and generalizable inference procedure that do not require retraining undirected graphical models are certainly of interest for a significant portion of the community. One important aspect of this work is that the generation of graphical models happens at the level of potentials therefore retaining the explainability of undirected graphical models. Moreover, the experiments are well thought off as they assess the generalization capability of AGMs inference procedure.

Weaknesses: The experimental protocol is unfortunately lacking.Moreover, this work fails to acknowledge other methods attacking the problem of generalizable inference at test time of undirected graphical models[1, 2, 3]. It would advantage the paper to separate themselves for the aforementioned work and even better compare to some of them. The scale of the experiments is too small. The paper only uses multi layered perceptrons, using convolutional neural networks and larger datasets (CIFAR-10 or SVHN for instance) would make the experiments on image data stronger. There are no experiments showing the advantage of generating potentials instead of data directly, this is important as [1, 2, 3] operate at the level of the data. Finally, a clear discussion of the computational costs will be helpful. [1] Ivanov, et al.. "Variational Autoencoder with Arbitrary Conditioning." International Conference on Learning Representations. 2018. [2] Douglas, et al. "A universal marginalizer for amortized inference in generative models." arXiv preprint arXiv:1711.00695 (2017). [3] Belghazi, et al. "Learning about an exponential amount of conditional distributions." Advances in Neural Information Processing Systems. 2019.

Correctness: The experimental results do not show error bars which makes it difficult to form a definite opinion about the results.

Clarity: The exposition is clear and the flow of the different sections logical.

Relation to Prior Work: The paper does a good surveying prior literature but misses a few relatively recent highly relevant literature. It will greatly benefit this work to differentiate itself from the aforementioned literature.

Reproducibility: Yes

Additional Feedback: *** score increased to 6 ***


Review 2

Summary and Contributions: In some inference scenarios we may not have access to the true graphical model, but we may have access to a distribution of plausible graphical models. This paper presents a new approach to perform inference on a distribution of graphical models. Parameters of an ensemble of graphical models are generated with Generative Adversarial Networks. Inference is then performed on these ensemble of graphical models. This allows for better generalization across a distribution of inference tasks where the graphical model is not fully defined for each particular task.

Strengths: The paper is very well written. The contribution is novel an well defined. The method is benchmarked over a wide variety of datasets.

Weaknesses: The main weakness I find is that the presented method (AGM) doesn’t outperform (EMG) with a very significant difference while it adds some extra complexity. Regardless of that, I still think the work is valuable to the community and even if the accuracy gap is “incremental”, the novelty of the algorithm is not.

Correctness: The claims and methodology seem correct to me.

Clarity: The paper is very well written and easy to read.

Relation to Prior Work: The work is properly contextualized. I think it may also intersect with meta-learning where an algorithm learns to be flexible on a variety of tasks i.e. learning to learn. It would be great if the authors add a small paragraph in the related work or conclusions relating their algorithm to this line of research.

Reproducibility: Yes

Additional Feedback: --- Post rebuttal update --- Most of the points raised by the other reviewers where addressed in the rebuttal, therefore my mark will remain as an Accept (7).


Review 3

Summary and Contributions: This paper is focused on learning models that are able to compute MAP inference on discrete variable problems where the set of conditioning variables may vary at test time. To model this, the authors propose a generative model over pairwise undirected probabilistic graphical model parameters titled AGM which is trained in an adversarial manner. A noise vector is transformed into a set of potentials using a neural network, from which marginals are obtained by using belief propagation; these marginals are then compared against the marginals corresponding to the training data using a WGAN loss. Experiments are run to evaluate this method on "inpainting tasks" (given some subset of variables, predict the states of the rest), where the distributions of which variables are presented and which must be predicted may vary between training and evaluation. There is also an evaluation of the generative modeling capabilities of this approach.

Strengths: This work is presented very clearly and is easy to follow. The idea of learning an ensemble of graphical models generatively in an adversarial manner is novel as far as I'm aware. The experiments are thorough in comparing the performance of AGM and GibbsNet in scenarios where the distribution of query variables changes during training and evaluation.

Weaknesses: I'm a bit confused about the evaluation of the approach. What is learned is a generative model over probabilistic graphical models; however, the focus in experiments I and II is on conditional MAP inference. In this setting, the model is being used as a structured output prediction model, and so comparisons are missing against other structured prediction models, examples being [1], [2], and [3] (see "related work" section for refs). [3] is of particular note, as it is also trained in an adversarial manner. If the primary use of this model is for conditional MAP inference, then it is important to understand how well AGM compares against other similar models. That being said, since the samples themselves are unconditional, this approach is at a disadvantage compared to these other approaches, which condition their "samples" on the input. I think the motivation is unclear here: why use an unconditional model to obtain graphical model parameters when we could use a conditional model to do so and train it on a variety of "inference tasks" so that it is robust to these? If the goal is primarily to be used as a generative model over discrete variables, then much more emphasis needs to be placed on this in the experiments section, and comparisons against more generative models needs to be made. The image datasets used are rather simple; it would have been nice to see experiments on a more complicated inpainting task, e.g. using semantic segmentations. Additionally, since inference is stochastic in nature, it is important to understand the variance in predictions made by using this method, and how this changes as you aggregate more samples. However, this is not presented or discussed in the experiments anywhere.

Correctness: The methods look correct, and the experimental evaluation seems sound. The title of the paper is highly misleading though - the inference isn't really learned, since it is using standard belief propagation techniques. There exist papers which are focused on implementing "inference networks" which are trained to approximate structured inference, and so reading the title may lead the reader to believe that a similar model is covered here.

Clarity: The paper is very easy to read and understand - you did a great job with this!

Relation to Prior Work: The relation to previous models is presented clearly; however, as discussed above, proper comparisons are not made against models trained for MAP inference tasks. Some additional references mentioned above: [1] Chen, Liang-Chieh, Alexander Schwing, Alan Yuille, and Raquel Urtasun. "Learning deep structured models." ICML 2015. [2] Belanger, David, and Andrew McCallum. "Structured prediction energy networks." ICML 2016. (there are a few followups to this paper as well) [3] Tu, Lifu, and Kevin Gimpel. "Learning Approximate Inference Networks for Structured Prediction." ICLR 2018.

Reproducibility: Yes

Additional Feedback: One additional comment: since tables represent average results over a few trials, it would be great to also have, e.g., standard deviations presented as well to get a sense for how consistent the methods perform. Overall, I think the work is interesting, but I think the focus of the experiments needs to be clearer. Whether the final focus ends up being on MAP inference, generative modeling of discrete variable problems, or some combination of the two, additional comparisons need to be made to appropriate models. ---------------------------------------------------------------------------------------------------------- POST AUTHOR RESPONSE UPDATE: Thanks for the clarifications you provided in your response. Due to these, as well as addressing my concerns regarding the datasets and variance of results, I am increasing my score. I still think the paper would be stronger if it included a comparison against some structured prediction approaches - even though they don't solve the broader problem described in the introduction, they can be applied to several of the tasks used during experimentation, and it would be interesting to see how everything compares.

[Author Response · NeurIPS 2020]

We thank the reviewers for their useful comments. We first clarify the minor confusion raised by **Reviewer 4** about the
focus of our approach (discriminative v/s generative). We then address all the individual reviewer recommendations.

**Essence of our work**: The purpose of our algorithm is to produce undirected graphical models *to perform inference*[1],
by *conditioning on any subset* of our random variables. We do *not* want to bake in any information about specific future
test inference tasks during training. It is true that when test inference tasks are known in advance, a model trained on a
mixture of those may outperform us[2] (**Reviewer 4**) or our model may be incrementally better (**Reviewer 3**). But how
well do we perform, compared to a model trained on the mixture, when we are both facing *a completely new task*?
Experiment II, our main experiment, now bolstered as described below, shows our superior generalization capabilities
to unseen inference tasks. Experiment III touches upon the generative capabilities of our model, such as the ability to
produce samples in one shot, only to show the added perks of choosing our method for *inference* in the first place.

**Expanding experiment II (reviewer baselines, larger data sets)**: All of [1],[2],[3] from **Reviewer 2** have now been
absorbed into related work. They allow conditioning on arbitrary subsets of variables, like us. However, being purely
neural, they require masks defined over the random variables during training, to match query patterns expected in
test inference tasks, but we are *completely agnostic to inference* during training. These models fit perfectly into our
experiment II setting. In table (a) below, we now use [1] under the `MIX` and `MIX-1` scenarios [3], under model name
VAEAC. As expected, `MIX` accuracies are high as the tasks were seen before, but accuracies of `MIX-1` fall drastically,
showing the comparative strength of AGM, which generalizes better to unseen tasks. The same is seen across data sets.
[1] was shown to be better than [2] in their paper and code for [3] is unavailable. GibbsNet (*with CONV layers*, as
requested by **Reviewer 2**) is also added to experiment II as baseline. Although inference-agnostic as us during training,
GibbsNet learns a *latent* space and is not resistant to corruption of pixels (c=0.5 task in table (a) below). The VAEAC
and GibbsNet baselines compare data-generating approaches to our potential-generating approach (**Reviewer 2**).

As shown in figure (b) below, experiment II now also includes the larger SVHN (**Reviewer 2**) and Stanford Background
Semantic Segmentation (**Reviewer 4**) data sets (+MNIST and Caltech-101). Metrics for each of these data sets are
reported in tables like (a). All our experiments show the trends seen in the original paper, but we did enough repeats to
*include error bars* (**Reviewer 2**, **Reviewer 4**) with maximum width of $\pm 0.5$.

**Other related work**: Models [1],[2],[3] mentioned by **Reviewer 4**, now added to our related work section, involve
graphical models like us, combined with neural modules, for inference. However, they assume a *fixed set of input and
output variables*, solving problems such as image tagging [1],[3] and classification [2], by learning: potentials in [1],
and energy functions in [2] and [3]. These methods do *not* solve our problem formulated in our paper introduction,
where we do *not* separate purely-input from purely-output variables, and we *permute* the identity of input and output
variable indices across data points. As an analogy, *one* of our models should be able to solve image tagging as in [1] or
[3], the inverse of that problem, as well as any inpainting pattern on the images. **Reviewer 3** rightly pointed out that the
idea of one model producing parameters for another, has its roots in meta-learning. We have consolidated the related
work section with: *Meta Networks [Munkhdalai, 2017]* and *Learning feed-forward one-shot learners [Bertinetto 2016]*.

**Additional analysis on method**: We add time and memory complexity of our method as requested by **Reviewer 2**,
relating the complexity of fully-parallelized belief propagation [Bixler, 2018] to edge set cardinalities induced by data,
and to the ensemble size used at test time. As requested by **Reviewer 4**, for *every* data set used in experiment I, we now
plot how accuracy, and variance of predictions changes with the number of samples (size of ensemble), in the appendix.

| MNIST | | | | | | |
|---|---|---|---|---|---|---|
| | | *Tested on* | | | | |
| *Model* | *Trained on* | f=0.5 | w=7 | c=0.5 | q=1 | mean |
| EGM | MIX | $93.6 \pm 0.2$ | $66.3 \pm 0.2$ | $82.6 \pm 0.2$ | $86.9 \pm 0.3$ | 82.4 |
| | MIX-1 | $87.4 \pm 0.1$ | $64.1 \pm 0.3$ | $68.2 \pm 0.1$ | $84.2 \pm 0.1$ | 76.0 |
| VAEAC | MIX | $94.2 \pm 0.4$ | $72.4 \pm 0.4$ | $79.8 \pm 0.4$ | $87.9 \pm 0.3$ | 83.6 |
| | MIX-1 | $85.5 \pm 0.4$ | $61.2 \pm 0.5$ | $65.1 \pm 0.4$ | $81.3 \pm 0.1$ | 73.3 |
| GibbsNet | - | $88.6 \pm 0.1$ | $70.5 \pm 0.2$ | $68.0 \pm 0.1$ | $87.1 \pm 0.1$ | 78.6 |
| AGM (Ours) | - | $95.5 \pm 0.1$ | $72.3 \pm 0.1$ | $79.2 \pm 0.2$ | $87.4 \pm 0.1$ | 83.6 |

(a) Updated table for experiment II, with baseline models: VAEAC and GibbsNet (with CONV). See footnote 3 for MIX and MIX-1 definitions.

(b) SVHN (top), Stanford Background semantic segmentation (bottom). Per row: Target (image 1), 70% query pixels (red) (image 2), output of AGM (image 3).

## Footnotes

[1] We agree with **Reviewer 4** for a title change to 'Training Ensembles of Discrete Undirected Graphical Models Adversarially, for Generalizable Inference', to avoid insinuating that we are learning inference algorithms.

[2] Note to **Reviewer 4**: indeed, our training procedure uses 'unconditioned' samples, but at test time, when answering one query, every graphical model in the ensemble *is conditioned on the same observed data*, as shown in figure 1(b) of the original paper.

[3] In table (a) above, `MIX` is a model trained on the whole mixture of tasks shown horizontally, while `MIX-1` is trained on all tasks but the one it is being tested on, to evaluate generalization to unseen tasks. Task definitions are given in the original paper.


[Meta-Review · NeurIPS 2020]

This paper presents a novel method for using undirected graphical models to perform inference on arbitrarily chosen subsets of random variables. Initial reviews all identified this as a novel and significant idea, but also raised several issues, mostly pertaining to the experimental validation. After author response and discussion, the reviewers feel these concerns were sufficiently addressed to recommend accepting this paper.